# A Thermal-Switchable Metamaterial Absorber Based on the Phase-Change Material of Vanadium Dioxide

**DOI:** 10.3390/nano12173000

**Published:** 2022-08-30

**Authors:** Zhongbao Wang, Yanli Ma, Ming Li, Liangfei Wu, Tiantian Guo, Yuejun Zheng, Qiang Chen, Yunqi Fu

**Affiliations:** 1The College of Electronic Science, National University of Defense Technology, Changsha 410073, China; 2Key Laboratory of Material Physics, Anhui Key Laboratory of Nanomaterials and Nanotechnology, Institute of Solid State Physics, Chinese Academy of Sciences, Hefei 230031, China; 3Science and Technology on Advanced Ceramic Fibers and Composites Key Laboratory, National University of Defense Technology, Changsha 410073, China

**Keywords:** metamaterial absorber, thermal-switchable, vanadium dioxide

## Abstract

This article presents a thermal-switchable metamaterial absorber (TSMA) based on the phase-change material of vanadium dioxide (VO_2_). VO_2_ thin film was deposited on sapphire substrate by magnetron sputtering followed by vacuum annealing treatment. Then, the prepared VO_2_ film was sliced into tiny chips for thermal-switchable elements. The surface structure of TSMA was realized by loading four VO_2_ chips into a square metallic loop. The absorption frequency of TSMA was located at 7.3 GHz at room temperature and switched to 6.8 GHz when the temperature was heated above the critical phase transition temperature of VO_2_. A VO_2_-based TSMA prototype was fabricated and measured to verify this design. The design is expected to be used in metasurface antennas, sensors, detectors, etc.

## 1. Introduction

Metamaterials are artificial composites that show abnormal physical properties and do not exist in the natural environment [1], and the design and application of metamaterials have become a research hotspot in recent years. The metasurfaces are a two-dimensional expansion of metamaterials, which inherit the properties of metamaterials and are more flexible in application. As a periodic metasurface with a characteristic loss structure, absorbing metasurfaces are widely used in electromagnetic stealth fields, which can absorb electromagnetic waves through the resonance of the surface structure and the loss of the dielectric layer [2]. Their absorption characteristics can be analyzed by an equivalent circuit [3]. Due to the complexity of the external environment, adjustable metasurfaces are especially used in multi-band switching and multi-functional regulation. The traditional adjustable metasurfaces mainly make use of the adjustable characteristic of the positive intrinsic-negative diode (PIN diode) [4], varactor diode [5], and MEMS tube to control the electromagnetic properties of the metasurfaces and to obtain different electromagnetic properties. However, it is still a challenge for these devices to be used at high frequencies. In the actual design process, the parasitic parameters of active adjustable devices and the limitation of working frequency will cause the test results of the design to be inconsistent with simulated results.

Various regulatory materials in nature are also used in the design of adjustable metasurfaces, such as transparent conductive oxides, ferrites, two-dimensional materials, and phase change materials. Under different external conditions, phase change materials have different electromagnetic characteristics [6]. For example, vanadium oxide (VO_2_) [7] and Ge_2_Sb_2_Te_5_ [8] are typical thermally induced phase change materials. They are usually used for optical device control, such as photoelectric switches [9] and smart window materials [10]. VO_2_ is one of the most studied switchable materials. It changes from the insulating state to the metallic state at ~68 °C. The phase change temperature of VO_2_ can be reduced to room temperature by tungsten doping [11]. Before and after the phase change of VO_2_, the change of resistivity can reach 2~5 orders of magnitude [12]. At the same time, the speed of phase change in VO_2_ can reach the nanosecond level [13]. At present, it is mainly used in optical [14] and terahertz [15] frequency bands. An infrared metamaterial absorber is designed by using VO_2_ with a cross structure to realize dual-band and single-band switchable functions [16]; a tunable reflected light metasurface is designed to realize continuous phase modulation of reflected light in the near-infrared wavelength range by electronic control [17]. The tunable terahertz resonator is designed by using the hybrid cross structure of the metal patch of VO_2_ to realize the tuning of single and double resonance modes [18]. A 90° twisted E-shaped resonator loaded with VO_2_ chips enables linear polarization conversion [19].

Compared with active devices working in the microwave band, VO_2_ is not affected by parasitic parameters or frequency in its working state. Moreover, VO_2_ can avoid the influence of feeder lines on the design. At present, VO_2_ is mainly used in adjustable antennas [20] and the switching design of some devices [21]. It is rarely used on the metasurface in the microwave band, except for one simulation on microwave absorption at multiple frequencies: multiple vanadium dioxide rings are used to realize multi resonance absorption [22].

In this paper, a thermally-switchable single frequency metamaterial absorber (TSMA) is designed based on the phase-change material of VO_2_. The theory of absorption is analyzed to guide the design of TSMA, and then the influence of the square resistance of VO_2_ chips and the size of the unit on the absorption effect is discussed. The simulation and experimental results show that when VO_2_ chips are in the insulating state, the metasurface realizes single frequency absorption at 6.8 GHz; after the phase change of VO_2_ chips, the absorption frequency moves to 7.3 GHz. The results of this study provide a valuable reference for the application of VO_2_ switchable metasurfaces in the microwave band.

## 2. The Theory of the Material Absorber

The traditional metamaterial absorbers generally consist of three layers: the top layer is a surface structure layer made of metal materials, showing periodic characteristics; the middle dielectric layer is mainly used as the loss layer; and the bottom layer is a metal ground to prevent the electromagnetic wave from penetrating. The metamaterial absorbers can be regarded as a dual-port network, as shown in Figure 1.

The reflection coefficient can be calculated as
(1)$$\begin{array}{c} \left|S_{11}\right|=\left|\frac{Z_{in}-Z_{0}}{Z_{in}+Z_{0}}\right| \end{array}$$
where $Z_{in}$ is the input impedance of this network, $Z_{0}$ is the characteristic impedance of the air.

The input impedance can be regarded as the parallel connection between the surface equivalent impedance $Z_{R}=R+jX=R+j\omega L+1/j\omega C$ and the equivalent impedance of the dielectric layer with the metal ground $Z_{d}$, which can be expressed as
(2)$$\begin{array}{c} Z_{in}=\frac{Z_{R}}{Z_{d}} \end{array}$$

When the thickness of the dielectric layer is d and the relative permittivity constant of the dielectric layer is $\varepsilon_{r}=\varepsilon_{r}^{\prime }+j\varepsilon_{r}^{\prime \prime }$, then $Z_{d}=jZ_{0}tan\left(\beta d\right)/\sqrt{\varepsilon_{r}}$.

Therefore, the absorption rate can be expressed as
(3)$$\begin{array}{c} A\left(\omega \right)=1-{\left|\frac{Z_{in}-Z_{0}}{Z_{in}+Z_{0}}\right|}^{2} \end{array}$$

In order to achieve a good absorption, a low reflection coefficient should be reasonably designed by optimizing the shape of the surface structure, the material of the dielectric layer, or the other parameters of the metasurface. The resonance and the equivalent surface impedance can be adjusted by changing the surface structure to improve the impedance matching effect between the absorber and free space. At the same time, the high dielectric loss can be realized by choosing an appropriate material with a specific thickness and the relative permittivity constant of the dielectric layer. In brief, to obtain a perfect material absorber, both the surface structure and the dielectric layer need to be designed reasonably.

## 3. The Design of the Metamaterial Absorber

The structure of the absorber is shown in Figure 2a. The surface structure consists of a square ring and four metal patches connected by four VO_2_ chips. The material of the dielectric layer is FR-4, with a dielectric constant of 4.3 and a loss tangent angle of 0.02.

The equivalent circuit model of the metasurface unit cell is shown in Figure 2b. The value of the equivalent capacitance $C_{r}$ in the metal ring is very small and can be ignored. When VO_2_ chips are in the low resistance state, there is an LC series resonance in the circuit, and the resonance frequency is $f=1/\left(2\pi \sqrt{LC}\right)$. When VO_2_ chips are in the high resistance state, it is in the disconnected state between the metal ring and the metal patch. The capacitance effect between the metal ring and the metal patches reduces the equivalent capacitance of the whole model. Compared with the low resistance state of VO_2_ chips, the resonant frequency will move to a high frequency. Therefore, this design can realize the reconfigurable function of switching the absorption frequency. The equivalent circuit simulation is carried out by the software ADS, and the full wave simulation results and circuit fitting results are shown in Figure 3 as follows. It can be observed that the circuit simulation results are in good agreement with the simulation results. The equivalent circuit parameters in the metal state are R=4.36 Ω, L=0.5 nH, C=317.48 fF. The equivalent circuit parameters in the insulating state are R=3.17 Ω, L=0.57 nH, C=350.85 fF.

### 3.1. The Preparation of Vanadium Dioxide

Full-wave simulation results are obtained using the commercial software package CST Microwave Studio 2020. Keeping the other conditions fixed, the relationship between the absorption performance of the absorber and the square resistance of VO_2_ chips is simulated, as shown in Figure 4. With the increase in the square resistance, the reflection coefficient becomes worse until it shows a total reflection state. Therefore, the smaller the square resistance after the phase change of the VO_2_ chips, the better the absorption effect.

VO_2_ thin film with expected resistance can be obtained by adjusting the material parameters. The square resistance of the thin film material $R_{□}$ is related to thickness, size, and resistivity. It can be calculated as
(4)$$\begin{array}{c} R_{□}=\frac{\rho }{h}=R\frac{w}{d} \end{array}$$
where ρ, h, d, and w are the resistivity, thickness, length, and width of the thin film material.

VO_2_ thin film was deposited on sapphire substrate by magnetron sputtering followed by vacuum annealing treatment. Before being put into the deposition chamber, the C-plane sapphire wafers were degreased ultrasonically using acetone and ethanol to remove the organic matter and other impurity ions that adhered to the surface of the substrates and were finally blown dry by nitrogen gas. The metal V target (purity: 99.99%) with a diameter of 3 inches was used as sputtering source material, and the distance between the target and substrate was about 200 mm.

The deposition chamber was evacuated to a vacuum better than 5.0 × 10^−8^ Torr before the sputtering gas was introduced. High purity Ar (99.999%) and O_2_ (99.999%) were used as working and reacting gas, respectively. The total gas pressure during reactive sputtering was changed from 0.1 to 0.5 Pa with the substrate kept at about 350, 425, and 475 °C during sputtering. After deposition, the films were annealed at 500~525 °C in a vacuum furnace of ~2 Pa. The detailed sputtering parameters can be found in Table 1.

According to Table 1, the thickness of the VO_2_ film increases with the increase in sputtering time. The annealing time has a great influence on the quality of the final film, and the annealing time required for the film produced under different conditions is not the same. After exploring the preparation conditions many times, the following preparation conditions were obtained: magnetron sputtering technology was used to sputter on a 0.2-mm-thick 2-inch sapphire substrate, which was heated to 425 °C, and the film was formed after 50 min of continuous sputtering. Next, after annealing at 525 °C for 3 h, the film thickness was measured with a step profiler, and the test result was 260 nm.

The crystalline structure was characterized by grazing angle X-ray diffraction (XRD) measurement achieved with a Philips X’Pert Pro MPD diffractometer, using Cu–Ka radiation with a wavelength of 1.5406 Å. The atomic force microscope (AFM, ParkSystemsNX10) was used to measure the surface roughness (Ra) of the VO_2_ film.

The surface morphology was characterized by the SEM, and the representative graph is shown in Figure 5a. Obviously, the VO_2_ films are composed of uniform nanoparticles less than 100 nm except for a few larger worm-like particles. As shown in Figure 5b, the XRD peaks are in accordance with the VO_2_ standard XRD spectrum (JCPDF Card 43-1051), indicating a high purity characteristic of VO_2_, and the surface roughness Ra of 8.6 nm was measured (the inset).

The multi-functional digital four-probe tester (FPT) is used to test the square resistance of VO_2_ thin film. The square resistance is 0.3 MΩ/□ under normal temperature and 56 Ω/□ after the phase change. The magnitude of the square resistance change of the VO_2_ film before and after phase change is about 3.73 orders of magnitude. Figure 6 shows the test environment and the change law of the square resistance. The phase change between the metal state and the insulating state of VO_2_ can be clearly seen in Figure 6. The temperature of the phase change is about 70 °C, and the width of the thermal hysteresis loop is about 15 °C.

### 3.2. The Design of the Material Absorber

The reflection coefficient of the absorber with different thicknesses of the dielectric layer is shown in Figure 7. When VO_2_ chips are in a metal state, the absorption effect increases with the increase in thickness. When VO_2_ chips are in the insulating state, the absorption effect decreases with the increase in thickness. For VO_2_ chips in different states, the designed absorber can be considered to have different surface structures. The thickness of the dielectric layer is set at 0.8 mm in consideration of absorption.

After optimization, the size of the unit cell is finally determined to be 9 mm. The outer length of the square ring in the surface structure is 6.5 mm, and the line width is 0.5 mm. The size of the metal patch around the square ring is 0.5 mm × 0.5 mm. The dimensions of the VO_2_ chip are 0.5 mm × 0.5 mm. We carried out modeling and simulation in the full-wave simulation software, and the absorption performances of the absorber under oblique incidence and different polarization were studied, as shown in Figure 8. In TE mode, the reflection coefficient decreases with the increase in the angle of the oblique incidence when the VO_2_ chips are in the metal state and is basically unchanged when the VO_2_ chips are in the insulation state. In TM mode, the reflection coefficient decreases with the increase in the angle of the oblique incidence when the VO_2_ chips are in the metal state and is basically unchanged when the VO_2_ chips are in the insulation state.

## 4. The Experimental Measure

In order to verify the correctness of the theoretical analysis and simulation results, we processed an absorber sample and tested it in the parallel plate waveguide. The sample contains 2 × 12 units with a size of 108 mm × 18 mm. The VO_2_ film is cut into 1.5 mm × 0.5 mm small chips by invisible cutting, which can be conducive to processing and manufacturing. The chips are bonded to the sample by silver paste, and the heating plates of the same size are bonded to the back of the sample to control the phase change of the VO_2_ chips.

When the upper plate of the parallel plate waveguide is uncovered, the upper plate contains two conical discs, which can generate vertically polarized transverse electromagnetic waves. Foam absorbers are filled around the waveguide to reduce multiple reflections at the edge of the waveguide. The test environment is shown in Figure 9. The test results of the parallel plate waveguide are recorded by a vector network analyzer (PNA Network Analyzer N5224B, KEYSIGHT), and the frequency test range is set to 6–8 GHz. We tested the samples at normal temperature and 100 °C.

Due to the limitation of experimental conditions, the absorption of the absorber is tested only in the case of vertical incidence. The test results are shown in Figure 10. The sample exhibits absorption of 16 dB at 7.1 GHz at room temperature; after the VO_2_ phase transition, the sample exhibits absorption of 8 dB at 6.3 GHz. The test results show that the sample has the function of frequency reconfigurability. Comparing the simulation results with the measured results, the difference between them is mainly due to the diffusion of silver paste on the surface of the VO_2_ chips during the process of bonding, which will cause the resistance of VO_2_ connected to the structure to change. According to the above simulation results, the different resistances between the sample square ring and the metal patch lead to different absorption depths. At the same time, during the processing, the silver paste also diffuses around the surface structure of the metasurface, which leads to the difference in the surface structure, resulting in the increase in the equivalent inductance. The increase in inductance can well explain the phenomenon that the resonant frequency shifts to low frequencies in the test results. Finally, the error of the metasurface size and the error of the parallel plate waveguide test system also lead to errors in the measurement. All of the above factors lead to the difference between the numerical results and experimental results.

## 5. Conclusions

In summary, a TSMA based on the phase change material of VO_2_ was proposed and investigated. The dual-frequency point of absorption was achieved by introducing four VO_2_ chips as active elements into the metal structure. The VO_2_ pieces were fabricated by magnetron sputtering and stealth cutting technology. The square resistance of the prepared VO_2_ thin film was 56 Ω/□ after the phase change, which was 3.65 orders of magnitude lower than that before the phase change. The high resistance ratio enabled VO_2_ to be used as a switchable thermal element. The TSMA showed different absorption frequencies in two states: the absorption located at 6.8 GHz and 7.3 GHz before and after the phase change of VO_2_, respectively. The test results are consistent with the simulation results, which verified that VO_2_ can be used in the metamaterial absorber in the microwave band. The design is expected to be used in metasurface antennas, sensors, detectors, etc.

## Figures and Tables

**Figure 1 nanomaterials-12-03000-f001:**
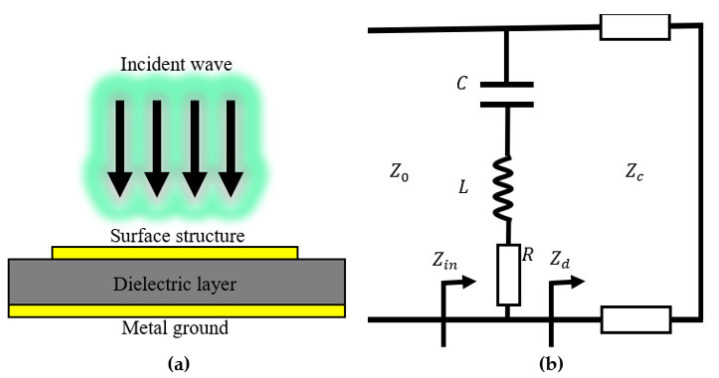
The model of the traditional metamaterial absorber (**a**) and the equivalent circuit (**b**).

**Figure 2 nanomaterials-12-03000-f002:**
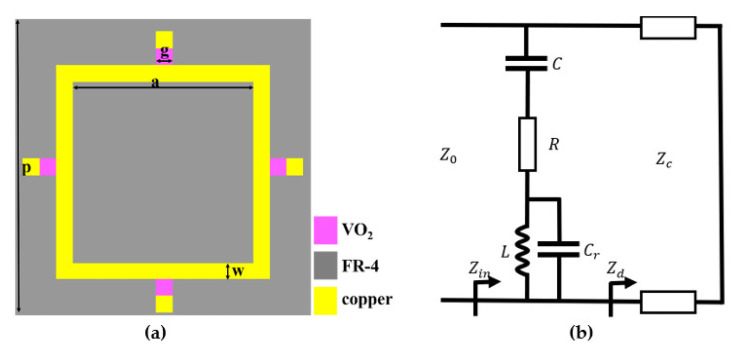
The model of the metamaterial absorber (**a**) and the equivalent circuit (**b**).

**Figure 3 nanomaterials-12-03000-f003:**
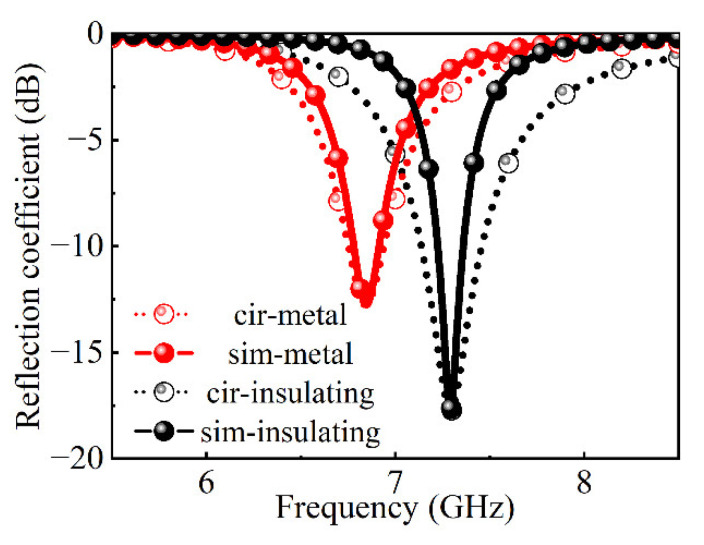
The circuit simulation and the full-wave simulation.

**Figure 4 nanomaterials-12-03000-f004:**
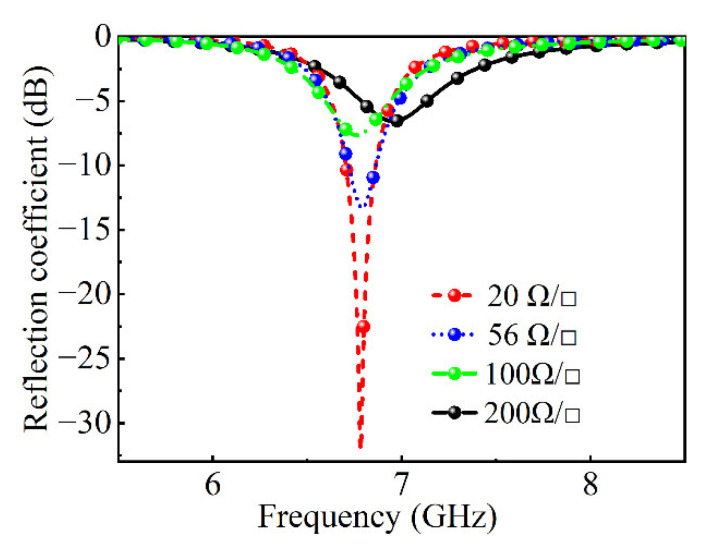
The reflection coefficient with the different square resistances of VO_2_.

**Figure 5 nanomaterials-12-03000-f005:**
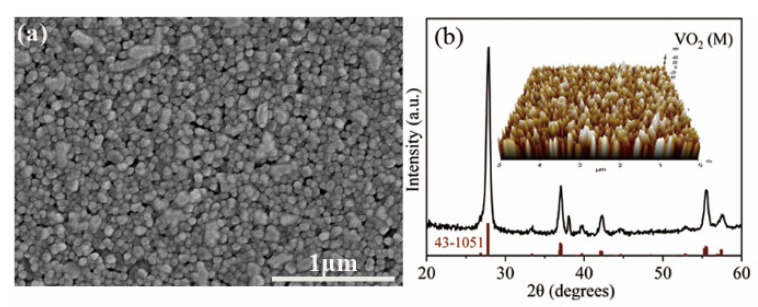
(**a**) SEM micrographs of the VO_2_ film deposited on the sapphire substrate; (**b**) XRD patterns of VO_2_ film (thickness: 260 nm), and the inset AFM image shows Ra = 8.6 nm measured by the software XEI Analysis with a scanning area of 5 μm × 5 μm.

**Figure 6 nanomaterials-12-03000-f006:**
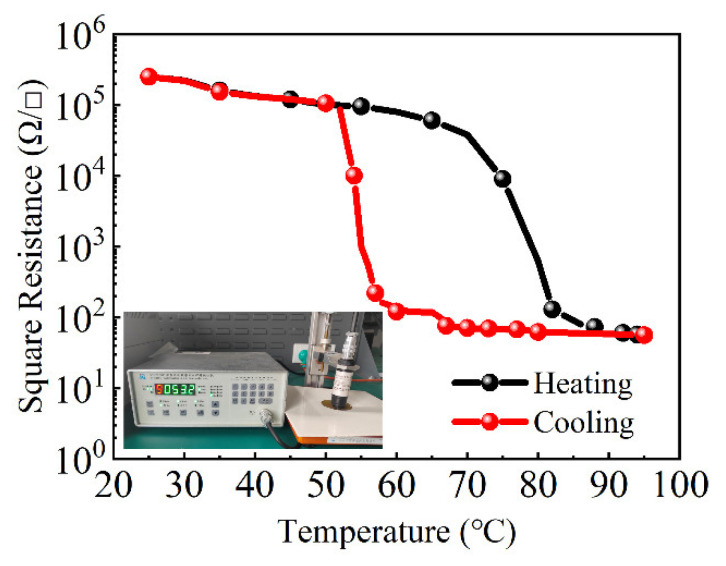
The square resistance of VO_2_ and the test environment.

**Figure 7 nanomaterials-12-03000-f007:**
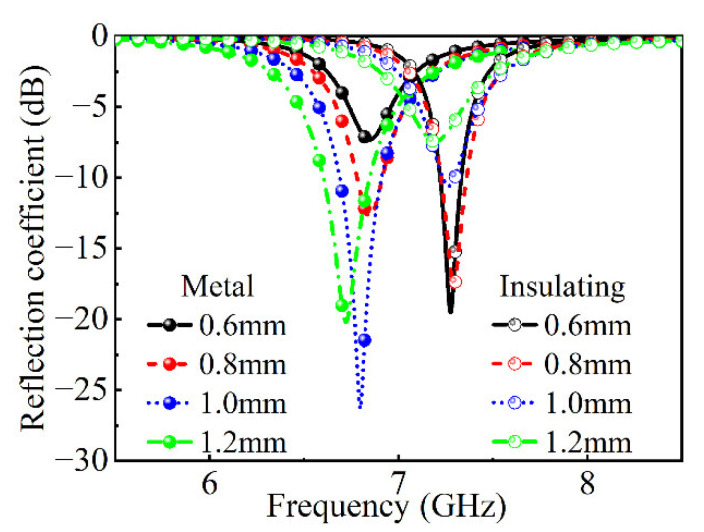
The reflection coefficient with the different thicknesses of the dielectric layer in the metal sate and in the insulating state.

**Figure 8 nanomaterials-12-03000-f008:**
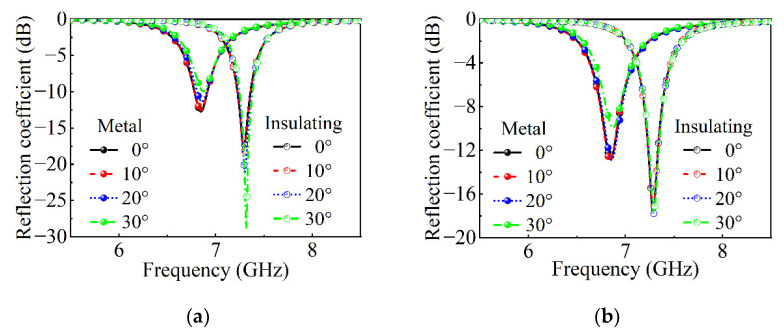
Full-wave simulation of the absorbing metasurface unit cell about the angle of incidence. (**a**) The reflection coefficient in TE mode in the metal state and the insulating state. (**b**) The reflection coefficient in TM mode in the metal state and the insulating state.

**Figure 9 nanomaterials-12-03000-f009:**
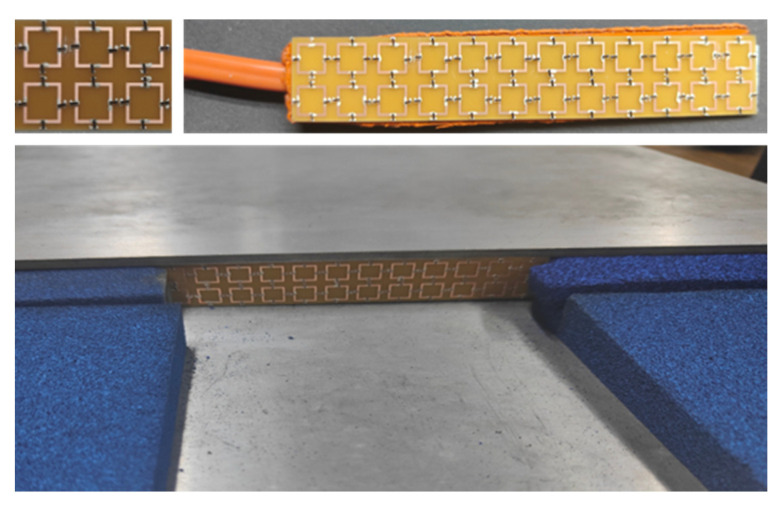
The TSAM unit, sample with the heating plate, and test device.

**Figure 10 nanomaterials-12-03000-f010:**
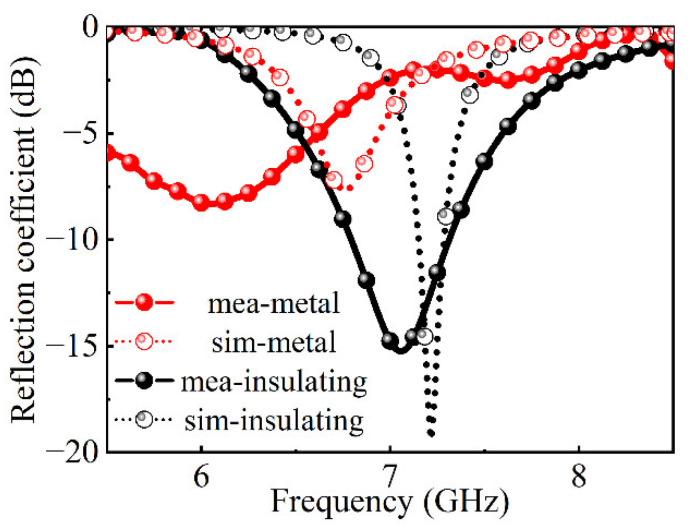
The test and simulation of TSMA.

**Table 1 nanomaterials-12-03000-t001:** The conditions and result of magnetron sputtering.

Substrate Temperature	Sputtering Time	Annealing Time	Thickness	The Square Resistance	
				Normal Temperature	Phase Change
475 °C	45 min	150 min	180 nm	0.5 MΩ/□	50 Ω/□
	60 min	150 min	210 nm	0.7 MΩ/□	2.8 kΩ/□
		180 min	210 nm	0.4 MΩ/□	190 Ω/□
425 °C	50 min	150 min	210 nm	0.5 MΩ/□	1 kΩ/□
		180 min	260 nm	0.2 MΩ/□	56 Ω/□
350 °C	40 min	150 min	330 nm	0.4 MΩ/□	150 Ω/□

## Data Availability

Not Applicable.

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
