# Peer review of "A Thermal-Switchable Metamaterial Absorber Based on the Phase-Change Material of Vanadium Dioxide"

_nanomaterials, 2022, doi:10.3390/nano12173000_

Round 1

Reviewer 1 Report

The authors reported a new design of thermal absorber based on vanadium dioxide as alternative phase-change material. The manuscript is well-written and the result is worthy of publication. However, some statements still require further justification. I think if the authors improved this manuscript it can be published.

1.    1)Im confused by using Argentum paste to fix VO2 elements into the absorber. Please, provide the information about thermal stability of ag-paste you used and be sure how big the resistance close to 100°C where sample were tested.

2.    2)It should be better to add measurements of resistivity zero-samples with Ag-bond without VO2. Cause the resistance of Ag will increase with increasing the temperature of experiment.  

3.    3)It well known about interaction VO2 and Ag. Diffusion process is slow, but change VO2 to rutile phase step by step. Could you please, give the information about reproducibility yout devise after couple cycles? How change the phase content of VO2 after 2-3 measurements?

4.    4)The actual thickness of VO2 layer 25% more than theoretical (set on magnetron). What method you use to confirm the thickness?

5.    5)It was mentioned the 70C is the phase temperature. Please, clarify, how big was the width of hysteretic loop, its usual characteristic to switcher.

6.6)Please, add concrete the working region (not only one wave/resistance/temperature or so) of thermal absorber and put it into conclusions and abstract.

Author Response

I should deliver my sincere gratitude to you on behalf of all authors. Your constructive and professional comments guarantee the high quality of a paper. Please refer to the attachment for the reply to your comment.

Reviewer 2 Report

The authors investigated VO2 loaded tunable metamaterial absorber oparating in microwave region.

The phase transition of VO2 resulted in absorption peak shift.

The authors discuss the deposition condition of VO2 and insident angle dependence of the absorption performance.

Experiments of the designed strudture was done and showed the proposed functionality.

I think the authors' effort on this work is great.

The detailed explanation about simulations and experiments is good point of this work.

However, basic points to be published in scientific journal are lucked.

Therefore, i can not suggest publication of this paper in the present form.

And, I suggest you proofread the English text.

List of points to be resolved

-First, there is insufficient research on previous studies.

Not many studies related to this study are cited.

Therefore, it is unclear what is the main novelty of this paper.

For example, [Scientific Reports,6, 23186 (2016)] etc. are not cited.

The number of references cited is also small.

-It is unclear why this metamaterial structure was chosen.

Is it designed to maximize the effects of VO2?

-Page 4, line 134, paragraph is not necessary.

-Is the VO2 quality higher than other studies?

For example, I suggest a quantitative comparison of FWHM of peaks and surface roughness.

-Regarding Figure 7, I was confused abouyt the definetion of the absorption.

Does deeper dip means high absorption?

If so,I think that the explanation about Fig. 7 is wrong.

-In the caption of Figure 8, both are TE.

-Regarding Figure 10, you only explain the difference between the calculation and the experimental results, and there is almost no discussion of the results obtained.

Author Response

(The authors gave the same response as above.)

Round 2

Reviewer 1 Report

Authors made additional experiments and improved the presented results. All my comments was fully answered. Manuscript can be published. 

Reviewer 2 Report

Thank you for your reply.

I think the authors adequately answer my comments and revised the manuscript.

Therefore, it is ready to publish in this journal.